# Vibrational noise disrupts *Nezara viridula* communication, irrespective of spectral overlap
Rok Janža[1,2], Nataša Stritih-Peljhan[1], Aleš Škorjanc[2], Jernej Polajnar [1] ✉ & Meta Virant-Doberlet[1]

Insects rely on substrate vibrations in numerous intra- and interspecific interactions. Yet, our knowledge of noise impact in this modality lags behind that in audition, limiting our understanding of how anthropogenic noise affects insect communities. Auditory research has linked impaired signal perception in noise (i.e., masking) to spectral overlap. We investigated the impact of noise with different spectral compositions on the vibrational communication of the stink bug *Nezara viridula*, examining courtship behaviour and signal representation by sensory neurons. We found negative effects of vibrational noise regardless of spectral overlap, challenging common expectations. Noise impaired the ability of males to recognize the female signal and localise its source: overlapping noise decreased sensitivity of receptor neurons to the signal and disrupted signal frequency encoding by phase-locking units, while non-overlapping noise only affected frequency encoding. Modelling neuronal spike triggering in sensory neurons linked disrupted frequency encoding to interference-induced alterations of the signal waveform. These alterations also affected time delays between signal arrivals to different legs, crucial for localisation. Our study thus unveils a new masking mechanism, potentially unique to insect vibrosensory systems. The findings highlight the higher vulnerability of vibration-mediated behaviour to noise, with implications for insect interactions in natural and anthropogenically altered environments.

Anthropogenic noise has become a ubiquitous feature of not only urban, but also natural environments[1,2]. The negative effects of noise produced by human activities on individuals and animal communities are now well recognized[2,3]; however, most studies have been so far focused on vertebrates[4,5]. While there is a growing awareness of the impact of anthropogenic noise on arthropods, the studies on arthropod taxa are still scarce[6–8]. Moreover, the effects of substrate-borne anthropogenic noise have only rarely been considered[9], even though most arthropods rely on substrate-borne vibrations in behaviours crucial for their reproduction and survival[10–12].

For arthropods, substrate vibrations are one of the most prevalent sources of information available in the environment, detected by highly sensitive vibrational receptors and used both in intraspecific communication and in predator-prey interactions[10–13]. The studies on the effects of vibrational noise so far mainly focused on impaired signal perception (i.e., masking) due to interference by other signals or abiotic noise caused by wind[14], consistently revealing negative effects[15–17]. In both laboratory and field conditions, animals avoided signalling in periods when vibrational noise could interfere with intraspecific communication, while potential spectral partitioning between signallers has rarely been observed[18,19]. Notably, all these insights were based on the use of noise spectrally overlapping the signals, inferred from air-borne sound communication where masking was considered to be restricted to such a situation[20].

Besides biological and geophysical components, anthropogenic vibrations are an integral part of the natural vibroscapes[19,21]. Vibroscape studies revealed a close frequency overlap between vibrations from biological sources and those produced by human activities[9,19]. The available information suggests that, due to the overlap in the frequency range below 2000 Hz, vibrational behaviour of arthropods may be especially vulnerable to interference by anthropogenic noise, and studies using synthesized white noise revealed negative effects on animal fitness and mating success[22–26]. Yet, our understanding of mechanisms underlying the observed effects is limited and the lack of such information does not allow us to reliably predict the long-term consequences of anthropogenic vibrational noise on arthropod communities.

[1]Department of Organisms and Ecosystems Research, National Institute of Biology, Večna pot 121, Ljubljana, Slovenia. [2]Department of Biology, Biotechnical Faculty, University of Ljubljana, Večna pot 111, Ljubljana, Slovenia. ✉e-mail: jernej.polajnar@nib.si

Here, we investigated the ability of the Southern green stink bug *Nezara viridula* (Hemiptera, Pentatomidae) to extract relevant information from signals used in sexual communication in the presence of vibrational noise. During courtship of this model species, the male must detect, recognise, and localise the stationary calling female on the host plant, a process mediated primarily by vibrational signals[27,28]. Signal recognition depends primarily on signal temporal parameters, while frequency composition may vary within the most sensitive range of vibroreceptors[29,30]. On the other hand, a delay between the arrival of vibrational waves to the receptor organs located in all six legs provides a crucial directional cue[31].

While previous studies on *N. viridula* have examined the behavioural effects of noise on these processes using narrow-band vibrations spectrally overlapping the signals[16,18], we utilised continuous white noise of different frequency bands and amplitudes to systematically investigate its effect on sensory processing and behaviour. Besides noise spectrally overlapping the signals, which in *N. viridula* contain most energy below 200 Hz[32,33], we also applied non-overlapping noise above this frequency range. While we expected that the former would impair recognition and localisation most strongly, we also hypothesised that these effects might extend beyond the range of spectral signal-to-noise overlap. In acoustic communication, behavioural effects of non-overlapping noise unrelated to masking – such as distraction and avoidance – are well-known[8]. However, signal masking by non-overlapping noise has also been documented[34]. To test these possibilities of interference in vibrational communication, we conducted behavioural assays on *N. viridula* males, assessing their ability to recognize and localise the female signals. Additionally, we investigated the representation of the signals in the leg nerves electrophysiologically and modelled a part of the effects we have discovered on signal encoding.

## Materials and methods
### Experimental animals
Adult *Nezara viridula* bugs were collected in autumn in the coastal part of Slovenia. In the laboratory, they underwent a 3-month hibernation followed by gradual activation at room temperature[16]. For the first two weeks of activation, the males were housed as a group, after which 30 males were randomly selected and isolated individually in plastic cups for an additional week before behavioural trials to reduce the influence of social experience on their behaviour. They were fed with fresh dwarf beans (*Phaseolus vulgaris* L.), peanuts (*Arachis hypogaea* L.) and sunflower seeds (*Helianthus annuus* L.).

*N. viridula* courtship behaviour involves a vibrational duet, prompting the male to search for the stationary, continuously signalling female. Both sexes use various low-frequency signals with the dominant frequency between 80 and 150 Hz, predominantly for calling and courtship purposes[35]. Vibrations are detected by a small number of leg vibrational sensilla, likely two in the subgenual organ (SGO) and six in the femoral chordotonal organ (FeCO)[36–38], altogether comprising three distinct physiological types.

### Stimulation
All experiments were conducted on a specialised anti-vibration table. Plant vibrations were induced using vibration exciters (Mini-Shaker Type 4810, Brüel & Kjaer, Denmark) placed on a thick layer of padding, which prevented any indirect 'leakage' of stimuli through the table to the plant. The exciters were driven by external sound cards (Sound Blaster X-Fi, Creative Labs, Singapore) and a power amplifier (PA1011, Rigol, China). These included a synthesized female calling song (FCS) train (in Audacity 2.2.2), comprising a sequence of 1.1 s long, 89 Hz sinus pulses with 0.1 s rise/fall time and a 2.8 s pause, mimicking natural song characteristics (see Supplementary Table S4). The stimuli evoked harmonic frequencies in plants, aligning with previous findings that signal harmonics may not be present at the source but result specifically from resonating plant tissues[33,39]. The song was played in a continuous loop. Band-limited white noise of three different frequency bands (Fig. 1) and three amplitudes relative to FCS (measured at the reference point on the plant; Fig. 2, point 7) was synthesised in Matlab (MathWorks, USA) based on an inverse fast Fourier transform[40]. The noise bands comprised the following: fundamental-overlapping noise (FON; 50-150 Hz) covering the fundamental FCS frequency, fundamental and harmonic-overlapping noise (FHON; 50–500 Hz) covering the entire FCS frequency range, and non-overlapping noise (NON; 500-1000 Hz) above the FCS frequency range. While the FCS amplitude was fixed, the noise RMS amplitude was adjusted to achieve a signal-to-noise ratio (SNR) of 6, 0 and -6 dB.

### Behavioural assays
**Plants and calibration.** We performed the experiment on six symmetrically shaped bean seedlings (*P. vulgaris* L., Etna variety) of similar size, replacing each plant with a fresh one every second day. Each plant was measured (Supplementary Table S2) and equipped with reflective foil at selected locations (Fig. 2). FCS was played from the left or right leaf via a vibration exciter, while noise was played back continuously from another exciter attached to the stem. The side from which FCS was played (i.e., ipsilateral to its application on the plant) was randomized for each trial. Vibrations were recorded using two laser vibrometers (PDV100, Polytec, Germany). Data were captured via a sound card (Sound Blaster X-Fi, Creative Labs, Singapore), using Raven Pro v1.5 (Cornell Lab of Ornithology, USA) and analysed using Microsoft Excel (Microsoft, USA) and R programming language (R Core Team, Austria). The experiments were video recorded (HC-VXF990, Panasonic, Japan) and subsequently analysed with VLC media player (VideoLAN, France).

**Fig. 1 | Playback stimuli.** Oscillograms (top) and spectrograms (bottom) of stimuli used in the experiments, recorded on the reference point of a plant (see Fig. 2). The noise was band-limited to three frequency bands (FON, 50–150 Hz; FHON, 50–500 Hz; NON, 500–1000 Hz; for abbreviations see Glossary, Supplementary Table S1). The expression of the female calling song (FCS) signals' harmonic frequencies on the plant varied, but in most cases, peaks above 400 Hz were undetectable.

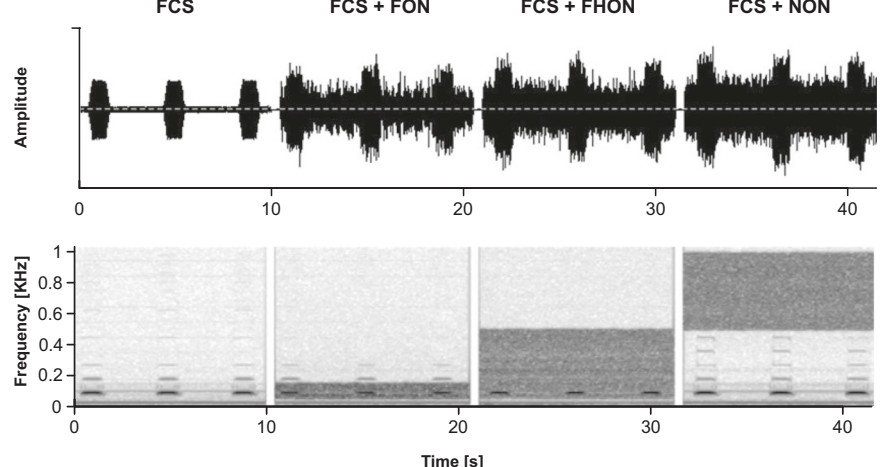

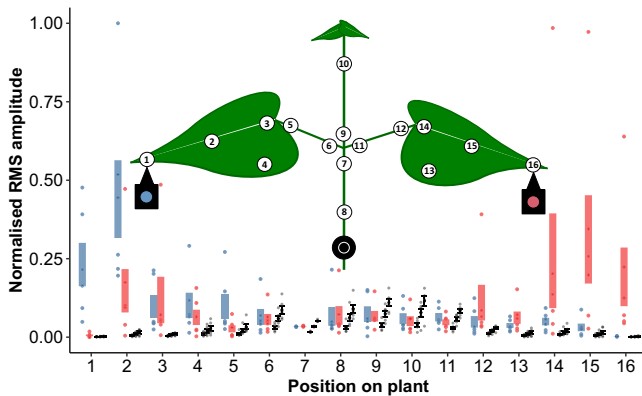

**Fig. 2 | A schematic representation of the behavioural experiment setup, with amplitude levels of FCS and FON registered at different positions on the plant.** A female calling song (FCS) was played back to either left (blue vibration exciter) or right leaf (red vibration exciter) while band-limited white noise was played back to the stem (black vibration exciter). The blue and red bars at each position represent the intervals between the mean +/- standard error of the mean for normalised amplitude of FCS played from the exciter marked with the same colour ($N = 6$). The black lines represent means with standard errors of three different amplitude levels of FON ($N = 6$; from left to right, 6, 0 and $-6$ dB SNR relative to FCS measured at the reference point, i.e. point 7; Supplementary Table S3). For other noise types, see supplementary Fig. S1.

Alongside the laser vibrometer at the reference point (Fig. 2, point 7), another one focused on the middle of the right leaf (Fig. 2, point 15) to monitor male vibrational signals.

The amplitude of synthesized FCS was adjusted to 0.3 mm/s RMS, which is within the natural range recorded in preliminary experiments from eight females signalling on a leaf close to the vibration exciter (Supplementary Table S5), but still allowing playback of noise at 6 dB SNR without distortion. Both FCS and noise were registered on the stem just below the stem-leaf branching (Fig. 2, point 7; i.e., the reference point) where males make directional decisions[31]. Due to filtering properties, the amplitudes of certain frequency components of noise were manually adjusted to ensure a flat frequency spectrum at this location (Fig. 1).

Signal and noise amplitudes were measured prior to the behavioural experiments at 16 locations on each of the six plants used. Measurements on the ipsilateral leaves consistently showed higher male signal amplitudes compared to applied noise, regardless of SNR or male position on the plant. While noise amplitudes on both leaves were much lower than the FCS amplitude, they matched or slightly exceeded the calling song amplitude on the stem and petioles (Fig. 2).

**Experimental design.** We utilised 30 males in a 12-day experiment (25 tested per day), with trials randomised by day and treatment. Each male underwent testing in nine noise and one control treatment, the latter with only FCS played back to the plant. At the start of a trial, the male was placed on top of the plant and was immediately subjected to simultaneous FCS and noise playback. Trials lasted 15 minutes from the male's emission of the first vibrational signal or start of searching (counting all walking movements as searching) or until the male located the FCS source or left the plant, whichever came first. We determined the percentage of signalling males, their cumulative signalling time, signal duty cycle, and latency to the first signal (the latter measured from the moment the male was placed on the plant) as indicators of female signal recognition (similar to e.g.[41–43]), and the percentage of males locating the FCS playback source (searching success) as indicators of their localisation ability. Cumulative searching distance was considered a context-dependent indicator: shorter distance in combination with lower searching success than control would indicate impaired signal recognition, while longer distance in combination with unaffected or decreased searching success would indicate unaffected signal recognition and

impaired localisation ability. If the male did not start signalling or searching in 15 minutes after being placed on the plant, the trial was ended and the male considered as non-participating; signalling and searching parameters were not calculated in this case.

### Sensory physiology

Experiments were performed on a specialised anti-vibration table and enclosed in a Faraday cage. Adult males were fixed dorsal side-up on a metal support, leg nerves were exposed and the preparation continuously perfused with Davenport saline. Tarsi of all legs were attached to the head of the vibration exciter using adhesive tape, with the femur-tibia angle at roughly 90 degrees. Legs were stimulated with the same combinations of signals and noise as in behavioural experiments. Likewise, the FCS amplitude was set to 1 mm/s root mean square (RMS), fixed across different treatments. Noise was applied at different amplitudes to match 24 to 0 dB SNR levels (in 6 dB steps). Extension to higher SNR levels than in behavioural experiments was used to determine the sensory threshold of noise effects, while the lowest SNR of -6dB was not tested due to excessive mechanical interference from the high-amplitude noise which masked the neuronal response. Stimulation consisted of three randomised groups of FCS-noise combinations played back to the males' legs. Each group started with 10 FCS signals without added noise, serving as control, followed by five series of FCS-noise combinations with increasing noise amplitude (24 to 0 dB SNR) for each noise type. Each series, preceded by a 10 s pause, started with 15 seconds of noise (allowing for neuronal adaptation) prior to FCS onset.

Details on the recording set-up and data analysis are provided in the Supplement. In short, spikes were recorded in the second and third leg nerves using a suction (Ag/AgCl2) electrode, and detected and sorted using Spike 2 (CED, Cambridge). A threshold curve was determined prior to the experiment in each preparation (see the supplementary Fig. S3 and accompanying details on stimulation) as a reference of the registered neuron activity. Timestamps of individual spikes were analysed in Matlab and R. Power spectral density and PSTH of neuronal activity were estimated using a Gaussian kernel convolution. In the power spectrum, the peak at 89 Hz represented FCS frequency and was normalised for comparisons between noise types and amplitude levels.

### Modelling

The model was based on the vibrational waveform (FCS overlapped by previously described noise types), recorded simultaneously on petioles of leaves ipsilateral and contralateral to the FCS source (Fig. 2, points 6 and 11), at the distance roughly corresponding to that between male legs stretching the junction (i.e., the directional decision point). SNR levels encompassed the whole range used in other experiments.

To simulate a sensory cell's response, we detected instances of waveform crossing a preset threshold, akin to spike triggering. To make the analysis more realistic, we introduced a 5 ms refractory post-spike period, which corresponds to the upper limit of phase-locking by *N. viridula* receptor neurons at 200 Hz[32]. In Matlab, we normalised the signal's RMS amplitude to 1, for a consistent threshold. We then identified events crossing this threshold and filtered out those occurring within the preceding event's refractory period.

To estimate the effect of noise on FCS frequency encoding, we measured time intervals between the threshold-crossing events. For each FCS-noise combination, we pooled data from ipsi- and contralateral petioles and plotted interval distributions (Fig. 6, upper row). We analysed event timing differences between petioles by calculating time delays. We first focused on ipsilateral events. We constructed intervals between the midpoints of these events and searched for contralateral events within the intervals. The time difference between the ipsilateral and the closest contralateral event within the interval was defined as the time-delay, with a positive value indicating that the ipsilateral event occurred first. We repeated this procedure for all contralateral events, excluding those already paired.

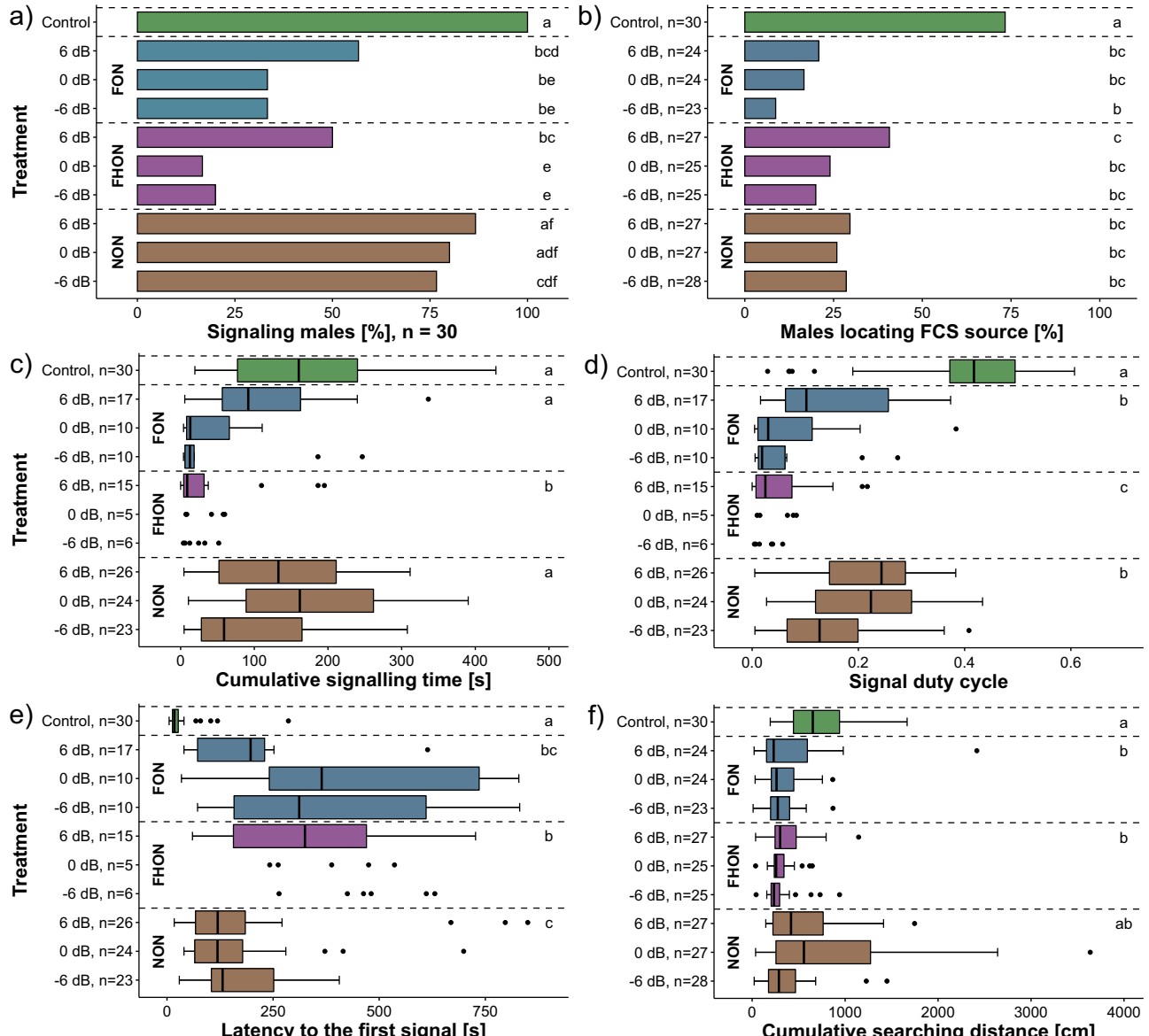

**Fig. 3 | The effect of noise on male signalling and searching. a** Percentage of signalling males, **b** percentage of males that located the signal source, **c** cumulative signalling duration, **d** signal duty cycle (cumulative signalling time/trial length), **e** latency to the first signal and (**f**) cumulative searching distance per SNR level. Statistical differences are denominated with letters adjacent to each plot. Due to the reduced sample size, comparisons with control were limited to the highest SNR level (6 dB SNR; *n* = 11 for signalling, *n* = 20 for searching). Plots a-b: Cochran's Q test and Pairwise exact McNemar test, *p* < 0.05. Plots c-f: Friedman test and pairwise Wilcoxon signed rank test, *p* < 0.05.

## Statistics and reproducibility

For statistical analysis of behavioural assays, we developed a custom R script. We used pairwise statistical tests to compare noise types and SNR levels, each individual undergoing ten treatments. Binary data (e.g., Fig. 3a, b) were analysed with Cochran's Q test (R package RVAide-Memoire v0.9-81-2) for differences between levels (*n* = 30). For the post hoc analysis, McNemar test (R package rcompanion v2.4.0) with FDR correction was used. Continuous numerical data (Fig. 3c–f) were analysed nonparametrically due to non-normal distribution, using the Friedman test (R package stats, v4.0.2) followed by the Wilcoxon signed-rank test (R package rstatix v0.7.2) with FDR correction. For continuous data, reduced male participation in trials due to noise limited comparisons to the control and the 6 dB SNR levels for each playback treatment (*n* = 11 for signalling, *n* = 20 for searching). We used a similar approach for the sensory physiology part of the study (*n* = 12), using the Wilcoxon signed-rank test with FDR correction to denote differences from the control (or 24 dB SNR in Fig. 5c) for the relative spike rate difference (Fig. 5a), spike rate during presentation of FCS in noise (Fig. 5b), spike rate during presentation of noise only (Fig. 5b), and the relative power of the FCS frequency in the spike train power spectrum (Fig. 5c).

To confirm the initial statistical analysis, we fitted linear mixed effects models from the "lme4" package in R[44]. For the behavioural parameters, we fitted the models using treatment as a fixed effect and male ID as random effect. For the sensory physiology parameters, we used SNR as a fixed effect and male ID as random effect. For all tested parameters, ANOVA showed that the effect of treatment was significant. We compared estimated marginal means for treatments with the Tukey method by using the R-package emmeans v1.10.3 (details in the Supplementary results).

In modelling the sensory cell's response, we utilized measurements from two distinct plants to emphasize substrate heterogeneity. Averaging distributions (see Figs. 6 and 7, and the supplementary material) across multiple plants would be less informative, since stimulus amplitudes varied

**Fig. 4 | Response of receptor neurons to stimulation.** PSTH diagrams of a representative recording of the summed response of receptor neurons during stimulation with FCS (control) or FCS with added noise, averaged over nine repetitions of the same stimulus. The noise was applied in three different frequency bands (FON, FHON, NON) at 6 amplitude levels, as in behavioural experiment; the signal-to-noise ratio (SNR) reflects changes in noise amplitude while keeping the signal amplitude constant. The mean spike rate value (red lines) and its standard deviation (pink bands) are shown. Rasters in the background represent individual spikes of the nine responses. Stimuli are shown at the bottom.

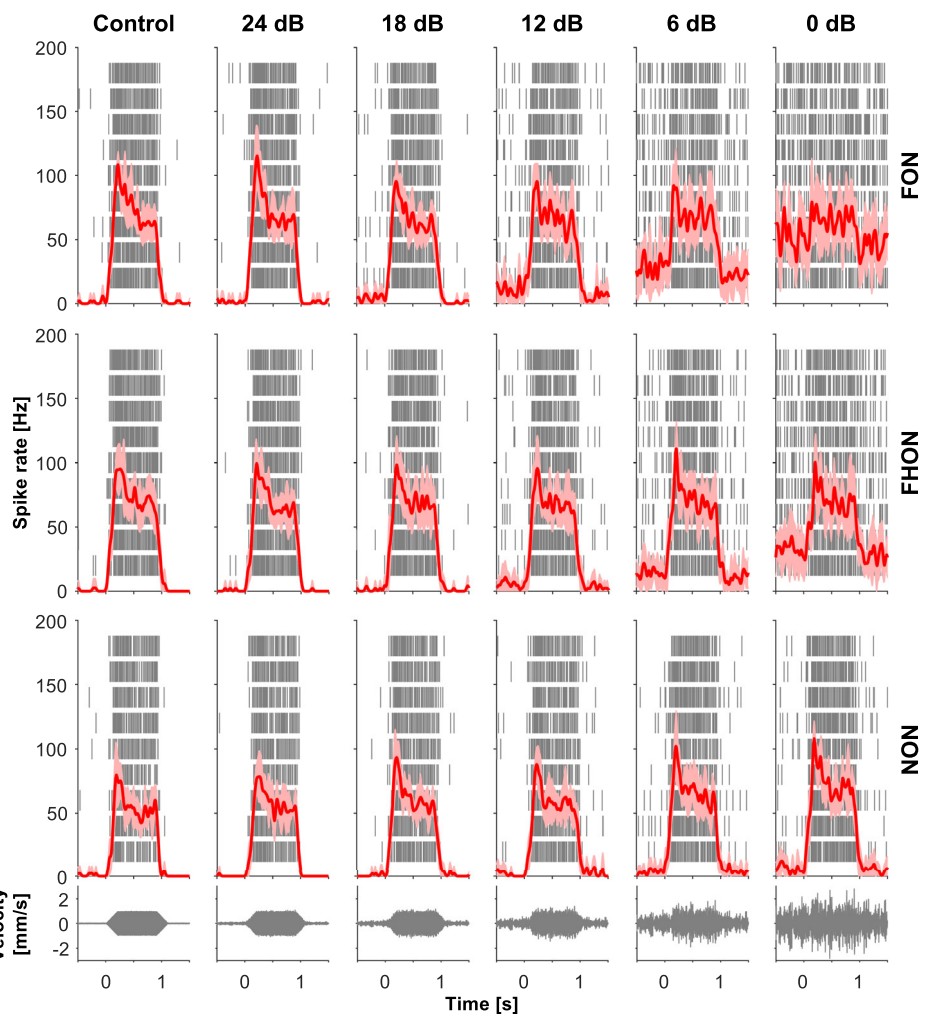

substantially between equivalent sections of different plants, and the locations of secondary peaks of the distributions differed as well.

### Reporting summary
Further information on research design is available in the Nature Portfolio Reporting Summary linked to this article.

## Results

### Noise impairs male courtship behaviour
Behavioural assays were used to quantify selected features of male courtship behaviour in the presence of noise, with various parameters of vibrational signalling and searching for the source of female vibrational signals (FCS) analysed to evaluate signal recognition and localisation ability.

Male behaviour was affected by noise of all tested bandwidths, with the greatest effect on both signalling and searching observed when noise overlapped the female signals (the fundamental overlapping noise (FON) and fundamental and harmonic overlapping noise (FHON) treatments (Fig. 3; see also Glossary, Supplementary Table S1). In both treatments, fewer males were signalling than in the control and the non-overlapping noise (NON) treatment [Cochran's Q test and Pairwise exact McNemar test, $p < 0.05$; details in supplementary Tabs. S6, S7] and exhibited shorter cumulative signalling duration and duty-cycle, especially evident in the FHON treatment. The latency to the first male signal was highest in FON and FHON treatments, with all these effects being strongest at the highest noise amplitude (-6 dB signal-to-noise ratio - SNR). Males' cumulative searching distance was lower in the FON and FHON than in the NON treatment, although the difference was not as prominent as seen for

signalling [Wilcoxon signed rank test, $p < 0.05$; details in Supplementary Table S8]. Localisation of FCS was less successful across all noise treatments compared with control, with different bandwidths and amplitudes having similar effect [$n = 30$, Cochran's Q test and Pairwise exact McNemar test, $p < 0.05$; details in supplementary Tabs. S6, S7]. The statistical analysis was confirmed by fitting linear mixed effects models to the data; details in supplementary Tabs. S9–S17.

### Noise impairs receptor neuron function
To elucidate the effect of noise on the detection of relevant substrate vibrations, we recorded the summed electrophysiological activity of the leg nerve. The sensory neurons responded to FCS by increasing the spike rate (Fig. 4). The initial peak in the rate was followed by a drop, indicating sensory adaptation. The sensory neurons also responded sensitively to the spectrally overlapping noise, with statistically significant increase in spike rate already at 12 dB SNR in both treatments (FON, FHON). The response to NON was weaker and increased significantly only at 6 dB SNR compared with the control (Fig. 5b, solid lines).

The response to the FCS, relative to the background neuronal activity, was only reduced in the presence of overlapping noise. This reduction was most prominent in the presence of FON, reaching significance at 12 dB SNR, while with FHON it was significant at 6 dB SNR (Fig. 5a). It mostly stemmed from increased response to noise alone (i.e., background activity), which was the strongest in the overlapping treatments (Fig. 5b). The effect was facilitated by an absolute response reduction with the overlapping noise, which may be ascribed to sensory adaptation (not being statistically significant, however; Fig. 5b, Wilcoxon signed rank test with FDR correction, $p > 0.05$).

**Fig. 5 | Summed response of receptor neurons to FCS at different SNR levels.** Mean ± SEM, n = 12 recordings. **a** Difference in spike rate between the response to FCS with added noise and noise alone (i.e., relative response). **b** Spike rate during presentation of noise alone (solid lines) or FCS with added noise (dashed lines). **c** Relative power of the FCS frequency (89 Hz, red arrow in the inset) in the spike train power spectrum. Note that a decreasing SNR reflects an increased noise amplitude relative to a fixed stimulus amplitude. **c** inset: average power spectral density of the spike train in response to FCS (control). Asterisks denote statistically significant differences from the respective control levels in **a**, **b** and from the respective 24 dB SNR levels in **c** [pairwise Wilcoxon signed rank test with FDR correction, $p < 0.05$].

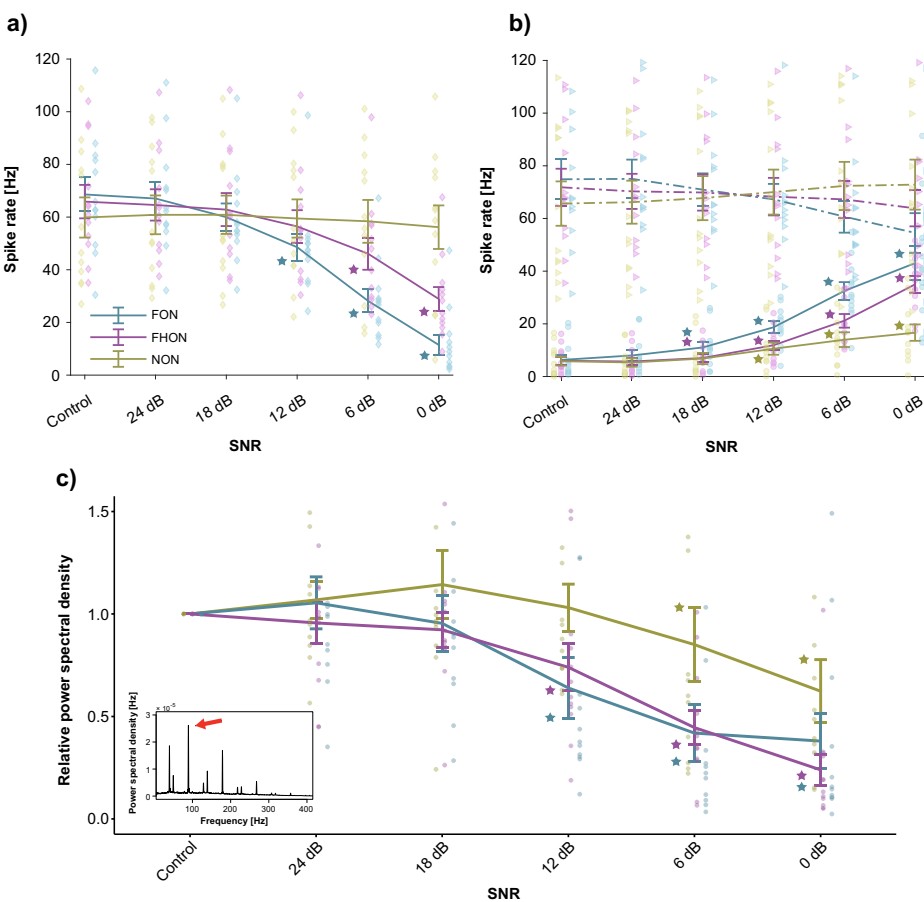

In the response to FCS with added NON, neither the absolute nor the relative response reduction was evident, even at higher amplitudes. The absolute response even increased, resulting from superposition of background activity (by a high-frequency receptor unit) onto the FCS response (by low-frequency receptor units).

Similar to the (relative) response to FCS evaluated as spike rate, the power of the FCS frequency component in the response also decreased notably, especially with the overlapping noise treatments. Specifically, the FCS frequency component in the response power spectrum was significantly reduced relative to 24 dB SNR already at 12 dB SNR in the overlapping noise treatments, while with NON, the reduction only became significant at 6 dB SNR (Fig. 5).

## Modelling the effects of noise on receptor neuron activity

FCS conveys information about the identity and the relative position of the female on the plant. While the presence is encoded in the spectro-temporal characteristics of vibrations, the position is determined using the delays between the sensory inputs to different legs[31]. To evaluate the effects of the noise on the encoding, we recorded vibrations on two opposing leaf petioles at a distance corresponding to the spacing between the bugs' legs. We identified instances of vibrations crossing a predefined threshold, akin to a hypothetical spike train of a sensory cell (for details see M&S). A raster plot demonstrates considerably distinct crossing patterns, as well as changes in time delays between ipsi- and contralateral crossings, depending on the amplitude and spectral properties of the noise (Fig. 6).

In the absence of noise, there was little variation in time intervals between the crossings, resulting in a narrow time interval distribution that was centred at the inverse value of the FCS fundamental frequency (Fig. 7a, grey patches). With low-amplitude noise, the distribution widened, indicating increased interval variability and thus breakdown of the FCS structure. This effect was observed for all noise types but at different amplitudes

(supplementary Fig. S4). As noise amplitude on the petioles increased, the interval distribution shape widened further, the main peak shifted, and secondary peaks emerged. In the case of overlapping noise treatments, the alteration occurred already at high SNRs (i.e., 12 dB; supplementary Fig. S4). However, with NON, the FCS period often persisted even at lower SNRs in the simulated neuronal activity, especially if the threshold was set just below the signal peak (supplementary Fig. S4). This was also reflected in the distributions of modelled neuronal responses (Fig. 7), revealing a consistent representation of FCS in the neuronal activity (the second peak in the two-tailed distribution). Note that the model represents the activity of a single sensory unit, and the neuronal data show a summed response of multiple receptor cells. Despite this, the two analyses are comparable as is evident from the effective representation of FCS period in the summed receptor response (Fig. 5c).

In the absence of noise, the distribution of time delays between the ipsi- and contralateral crossings was narrow, with the signal always reaching the ipsilateral point first (Fig. 7b). With increasing noise amplitude, the distributions widened and the main peak shifted in some cases into negative values, indicating that the signal reached the contralateral point first (Supplementary Fig. S5). As with time interval distributions, this effect was seen in all noise types but was more prominent with non-overlapping noise.

In addition to noise, transmission properties of the plant also influenced the distribution of time intervals and delays between the ipsi- and contralateral points (comparison between the two plants for all tested amplitudes is shown in Supplementary Fig. S5).

## Discussion

Our study explores the influence of band-limited white noise of different spectral compositions on vibration-mediated courtship in the stink bug *N. viridula*, revealing its negative effects both when overlapping and when outside the signals' frequency range. Field recordings showed that on plants

**Fig. 6 | Modelled neuronal response to substrate vibrations.** The FCS with added noise, recorded on the ipsilateral (black, Fig. 2, point 6) and contralateral (red, Fig. 2, point 11) petiole of a bean plant (oscillograms; lower diagrams) with modelled neuronal response of a hypothetical phase-locking sensory neuron (raster plots above the oscillograms). Raster plots show crossings of 1×RMS amplitude (normalised to 1) with a 5 ms refractory period. FCS was emitted to the leaf of the ipsilateral petiole and noise (FON, FHON, NON) was emitted to the bottom of the stem (Fig. 2, black vibration exciter). SNR at the reference point (Fig. 2, point 7) is shown at the top in bold, while SNR measured on the ipsilateral petiole is shown above each subplot.

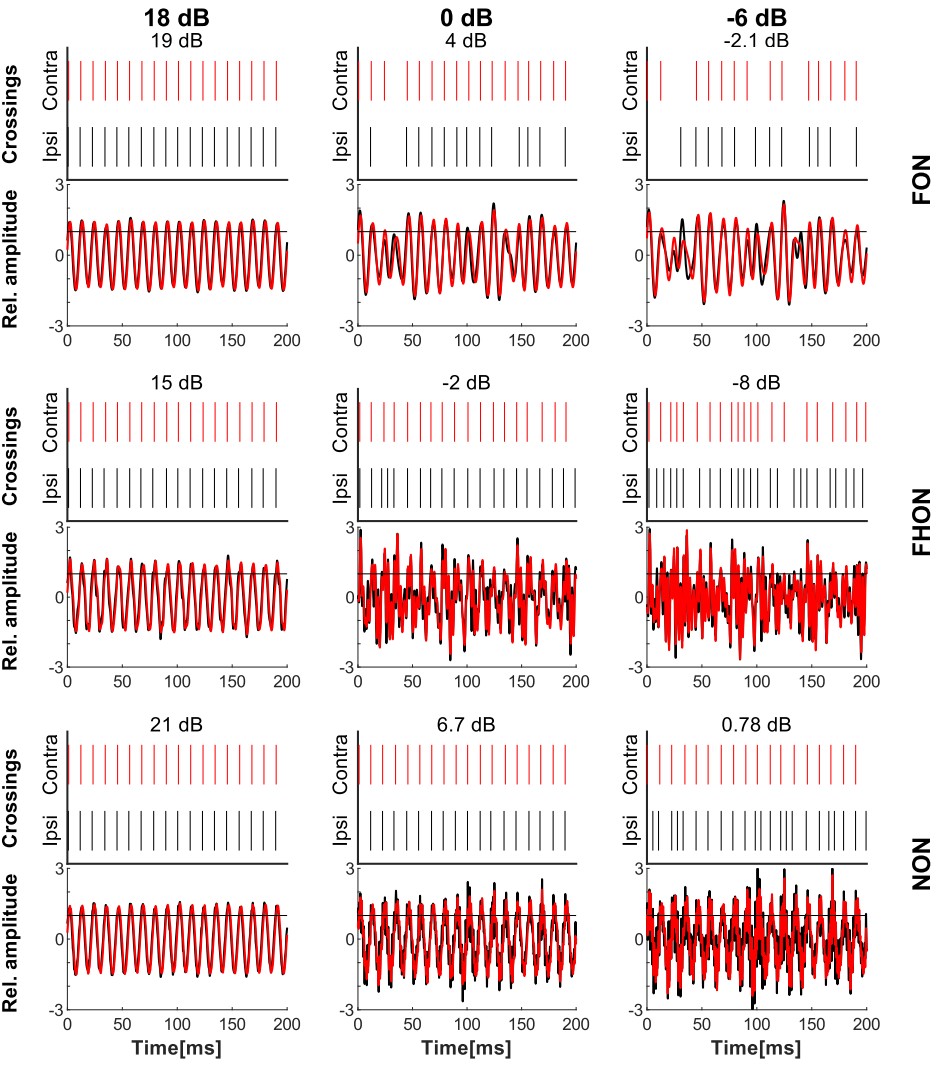

most energy of insect vibrational signals is focused in the frequency range below 1000 Hz and spectrally overlapping vibrations originate from geo-physical and biological sources[19,21], while anthropogenic vibrations extend also to higher frequencies[21,45]. These diverse sources of noise, encompassing frequencies up to several kHz, represent real-world conditions that could potentially disrupt vibrational communication.

We found that all applied noise types impaired in males both the ability of recognition of the female signal and the ability to localise its source. While the negative effect on signal recognition was the strongest in the presence of spectrally overlapping noise (see also[16,30]), source localisation was affected to a similar degree by all noise treatments. Notably, we demonstrated the corresponding noise effects on the receptor neurons' encoding of signal identity and location (Fig. 8).

These findings challenge the conventional view from bioacoustics, which relates masking as a process of impaired signal perception to cases of spectral (and temporal) overlap between the signal and the disturbance[20,46,47]. Based on this premise, the influence of non-overlapping noise on communication has been largely overlooked and studied almost exclusively from the view of other consequences, such as behavioural distraction and noise avoidance[8,48–50]. In many cases, however, the specific mechanisms underlying the effect have not been explored, like, for example, in the impairment of cricket phonotaxis by anthropogenic noise[51]. Recently, a study of bat echolocation addressed both the behavioural and neuronal levels of the disturbance, demonstrating that non-overlapping acoustic noise can also effectively mask the signals[34]. Similarly, our examination of

insect vibrational communication provides another piece of evidence for compromised signal perception caused by spectrally non-overlapping noise. While in bat audition the observed effect has been associated with processing mechanisms in the central nervous system (referred to as "informational masking"[47]), our findings in the stink bug suggest a physical interference between the signal and noise, impairing signal detection at the sensory organ level (referred to as "energetic masking"[52]), as is discussed in more detail below.

We found that male signalling and searching, reflecting the correct identification of the female signal[41–43], were hindered most strongly by noise spectrally overlapping the signal. Only these noise treatments led both to reduced spike rate (i.e., reduced sensitivity) in the response to the signal, consistent with the classical masking mechanism[20], and the disrupted signal encoding in phase-locked neuronal responses (discussed in detail below). The strongest disruption in the identification-related behaviour occurred when noise overlapped both the dominant and higher harmonic frequencies of the female signal[32]. This particular noise type should most strongly reduce the activity of middle-frequency receptor neurons (MFR), the only known leg receptor unit tuned to the signal's second harmonics[32]. Although the summed neuronal response to the signal did not explicitly illustrate this, as it mainly reflects the activity of predominantly low-frequency receptor neurons (LFR) in the legs of stink bugs[32,38], the reduction in MFR response provides the most likely cause of the observed behavioural effects. In *N. viridula*, the significance of detecting both dominant and higher harmonic frequencies of the female signal for its identification aligns with previous

**Fig. 7 | Distributions of modelled neuronal responses to substrate vibrations.** Distributions of (**a**) intervals between modelled vibration threshold crossings and (**b**) time delays between modelled ipsi- and contralateral threshold crossings in two spatially separated legs. Data for FCS with 6 dB (top) and -6 dB SNR (bottom) and three thresholds (0.0, 0.5 and 1.0 RMS) are shown. The actual SNR value on the ipsilateral petiole is shown above each inset. Reference distributions for FCS without added noise and 0.0 RMS threshold are depicted as grey patches (top out of scale). Other noise amplitudes and data for the second plant are shown in the supplementary Fig. S4 (intervals) and S5 (time delays), and examples of vibration oscillograms and 1.0 RMS threshold crossings are shown in Fig. 6.

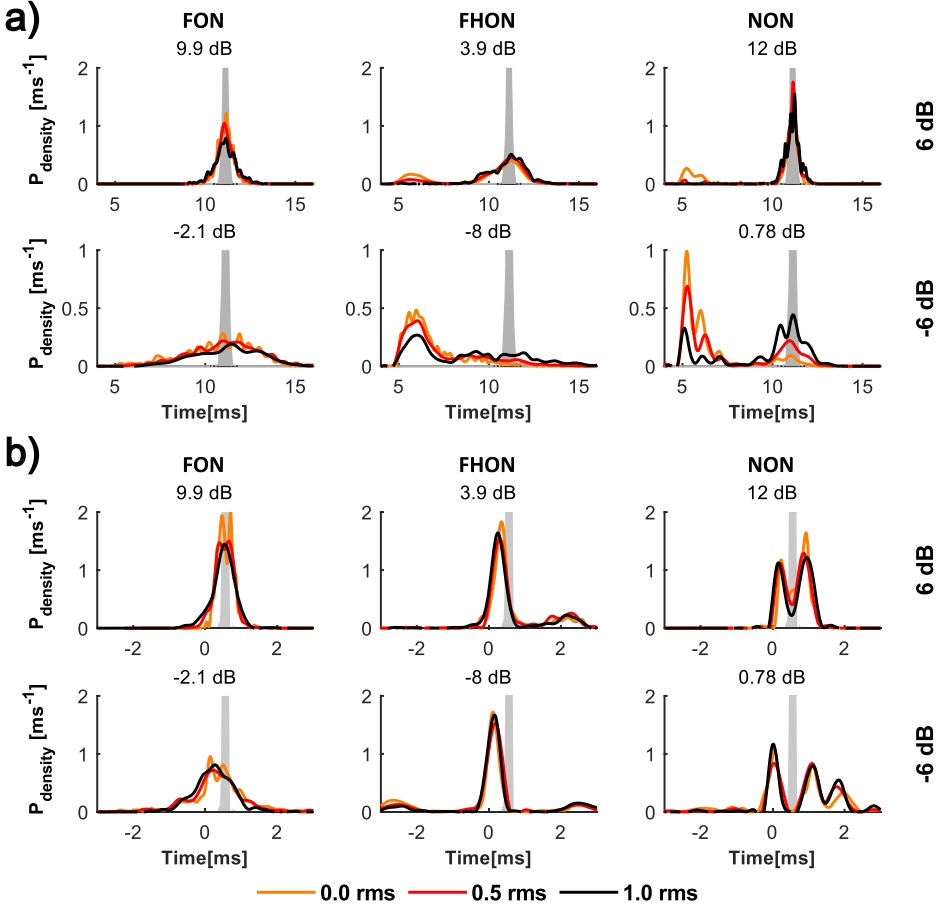

**Fig. 8 | Results of all three avenues of research – a schematic summary.** All tested SNR levels are presented in the same scale on the graphs' X axes, and the Y axes show the reduction of parameters related to recognition or localisation compared with the control without noise playback. For behavioural and neurophysiological experiments, ranked statistical differences from control are shown, while power density at and outside the reference distribution of spike intervals (for recognition) and the reference distribution of delays (for localisation) are shown for the neuronal model.

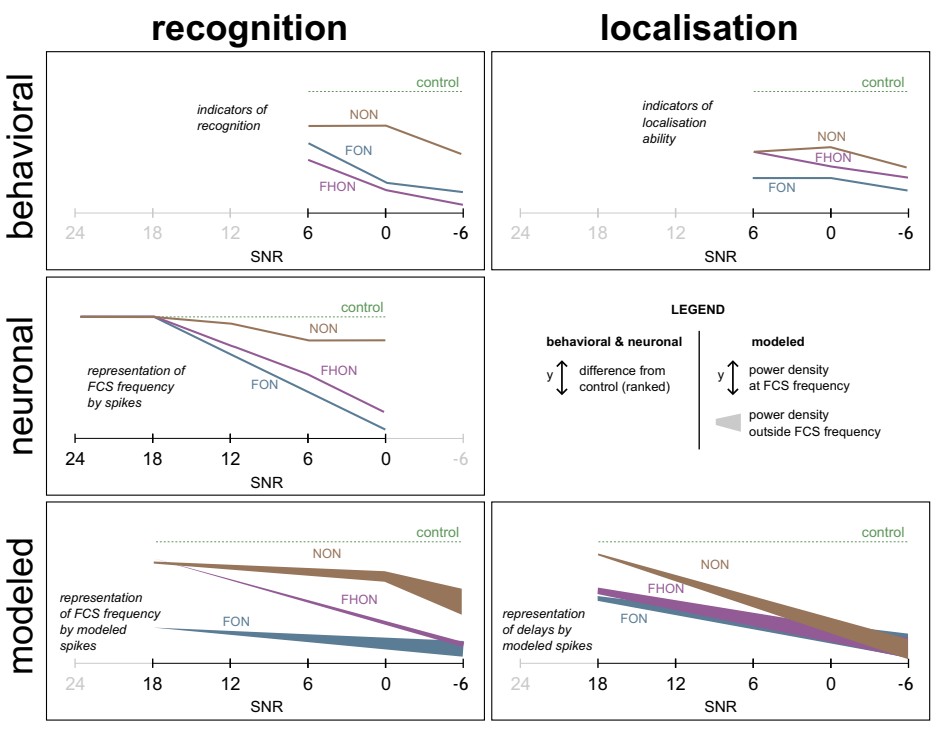

findings in other Hemipteran species, such as the planthopper *Hyalesthes obsoletus*[43] and leafhopper *Homalodisca vitripennis*[53].

In the vibrosensory system of insects, signal frequency is generally encoded not only in the neuronal typology[38,54], like in the auditory systems of insects and vertebrates[55,56], but also directly in the spike discharge when synchronised with a specific phase of the vibratory stimulus[13,54]. In *N. viridula*, such phase-locked responses occur in the frequency range up to 200 Hz in the LFR and the MFR neurons[32]. We found that both overlapping and non-overlapping noise disrupted this frequency encoding through distortion of the signal waveform, as evidenced through the reduced power spectrum peak at FCS frequency of the summed neuronal response, and the distorted spike triggering predicted by the model of a phase-locking neuron.

In the case of overlapping noise, this disruption of frequency information can be attributed to the summed effect of decreased sensitivity to the signal and distortion of the signal waveform. In the case of non-overlapping noise, which does not affect neuronal spike rate in response to the signal, alteration of the signal waveform provides the most direct explanation of the disrupted behaviour. While the interference with signal frequency information by non-overlapping noise is weaker than that by overlapping noise, its statistical significance at the SNR of 6 dB matches its behavioural effect. This highlights the crucial importance of accurately encoding the female signal's dominant frequency for proper identification. While the signal waveform alteration by the non-overlapping noise had only a minor effect on the proportion of males replying to the female, consistent with previous findings of a broad effective range of the female signal frequency in that respect[30], we revealed its subtler influence on signal recognition. This is evidenced by the increased response latency and decreased signalling time of males, indicating the necessity of extended neuronal processing before generating a response.

During search for the calling partner on the plant, *N. viridula* males typically extend their legs across different possible paths of an encountered branching before taking a directional decision[31,57]. Due to the irregular fluctuation of signal amplitude with distance on plant substrates, the delay between the arrival of the vibrational wave to receptors in different legs provides the only reliable directional cue[31,39]. Vibratory interneurons of this species encode delays as short as 0.5 ms in the latency to the first spike[31]. Our model demonstrated that this information encoded by receptor neurons becomes highly unreliable when either noise type is added to the signal, underscoring the significance of an unaltered signal waveform for its successful localisation. This was not as evident with the overlapping noise, where the male's reduced ability to localise the female signal partially stems from reduced signal identification, hindering the search process. However, in the presence of non-overlapping noise, where male searching remained largely unaffected, disrupted localisation can be attributed entirely to the distortion of the signal waveform. This distortion represents the synergistic effect of noise and complex transmission of vibrations through the non-linear plant medium.

Generally, we observed significant negative behavioural effects of noise already at an SNR of 6 dB, along with a corresponding reduction in sensory response to the female signal. In case of overlapping noise, signal frequency encoding became impaired already at the SNR of 12 dB. These data reveal a high susceptibility of vibrational communication in *N. viridula* to noise interference (see also[18,58]), similar to that found in other plant-dwelling Hemiptera, where noise of the same amplitude as the signals disrupted communication almost completely[24,59]. This contrasts the performance of insect acoustic communication in noise: *Mecopoda elongata* bushcrickets communicate nearly unaffected in the presence of continuous, spectrally similar signals of a sympatric species, down to the SNRs of -8 dB even when the signals are broadcast from the same direction[60]. This difference between the systems stems from a much lower capacity for vibrational frequency discrimination by Hemiptera (and other insects) compared with the elaborate hearing of the Ensifera. While insect vibrosensory organs detect only a few distinct frequency bands (regardless of the organ complexity)[13], ensiferan auditory organs possess numerous sensilla, each tuned to only a small part of the hearing

range[61], ensuring that even a minor spectral difference between signal and masker can enable signal detection[60,62]. Because of the relatively broad filter properties of vibrational sensilla, by contrast, the "critical band"[20] of vibratory noise impairing signal detection is generally much broader.

In the Ensifera, the efficiency of resolving auditory signals from noise is further notably improved when the sounds are received from different directions[63,64]. This phenomenon, called the "spatial release from masking", is of limited effectiveness in vibrational communication, however, especially on plant substrates[58]. We found, accordingly, that both behaviour of the males exposed to spatially separated noise and signal sources on the plant and the encoding of the female signal in their receptor neurons in an isolated leg nerve were impaired at the same noise levels. This implies not only the lack of effectiveness in the spatial release from masking but also in other central processing mechanisms aimed at enhancing communication in noise, such as gain control for suppressing weaker competitive signals in insect audition[65,66].

To conclude, insects are essential to ecosystem functioning[67,68], yet their populations are declining due to human activity[69,70], with the overlooked impact of noise pollution[6–8]. Despite the widespread use of vibrational signalling in insects[10,12], understanding of its basic features and susceptibility to noise remains limited.

The available information suggests that due to the spectral overlap of vibrational signals and anthropogenic noise in the range below 2 kHz[9,19], vibrational behaviour may be especially vulnerable to interference by anthropogenic noise. In this narrow frequency range of insect vibrational signals, constrained by filtering properties of their resident plants[10], a shift in signal frequency to avoid masking from anthropogenic noise, as described in insect air-borne sound communication[71,72], is severely limited. Recent field studies have also shown that airborne sounds may also be reflected in the vibrational channel to a notable extent both below and above 2 kHz[19,73]. Thus, direct impact of sound pollution could extend to many animal species and species assemblages that rely on substrate-borne vibrations and are not commonly considered vulnerable[74].

In addition, we demonstrated direct physical (energetic) masking of vibrational signals by noise outside the frequency range relevant to communication of our study species, challenging common expectations. We present a previously unexplored effect of signal and noise interference, which may be specific to insect vibrational systems due to their specificities of information encoding. Finding impaired communication at high SNRs implies that such effects of continuous noise on insect communities might extend over considerable distances from the noise source. Importantly, since noise-impaired orientation, not only mate finding but also predator-prey interactions, are likely to be directly affected by anthropogenic noise. These findings highlight the need for better understanding of the complexity and mechanisms of vibrational noise effects, as a prerequisite for effective conservation of neglected animal communities in our changing world.

## Data availability
All information needed to reproduce the results of the paper is in the paper and the Supplementary Materials. Raw data extracted from audio and video recordings, and electrophysiological recordings are archived at the Zenodo data repository[75] (https://doi.org/10.5281/zenodo.11910334). Due to the large file sizes, original audio and video recordings are available per reasonable request to the authors.

## Code availability
Matlab and R code for reproducing the analysis are archived at the Zenodo data repository along with raw data[75] (https://doi.org/10.5281/zenodo.11910334).

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

## Acknowledgements

The authors thank Dr. Andrej Blejec and Urban Dajčman for statistical advice. Additionally, Dr. Rok Šturm and Dr. Anka Kuhelj contributed valuable suggestions during the planning of behavioural experiments. We acknowledge funding by the Slovenian Research and Innovation Agency (ARIS) through the core research funding programme "Communities, interactions and communications in ecosystems" (P1-0255), the research project "Vibroscape: discovering an overlooked world of vibrational communication" (J1-3016) and PhD scholarship to RJ within the Young Researcher programme, all awarded to the National Institute of Biology.

## Author contributions

Conceptualization: R.J., N.S.P., J.P., M.V.D.; methodology: R.J., N.S.P., A.Š., J.P.; experimental work: R.J.; data analysis: R.J., A.Š.; writing-draft: R.J., N.S.P., M.V.D., writing-review and editing: R.J., N.S.P., A.Š., J.P., M.V.D.

## Competing interests

The authors declare no competing interests.
