## [Transparent Peer Review file · Communications Biology]

Vibrational noise disrupts *Nezara viridula* communication, irrespective of spectral overlap

Corresponding Author: Dr Jernej Polajnar

Version 0:

Reviewer comments:

Reviewer #1

(Remarks to the Author)

This is a very well written and comprehensive paper on the impact of vibratory noise on female-male communication in the stink bug *Nezara viridula*. The experimental design was clever and rigorous and the results seem clear. I applaud the authors on combining behavioral experiments with electrophysiology and modelling, as they are able to tell a very complete story. I only have a few comments that I hope might improve the manuscript further.

Specific Comments

(1) In reading recent papers on this topic, I found it noteworthy that the vibroscape can encompass both substrate-borne, but also airborne sound (e.g., Choi et al., *Communication Biology* 2024; Pessman et al., *Entomologia Experimentalis et Applicata* 2024). This seems especially relevant to the story that is being told here about the impact of non-overlapping noise. If there is space, it would be good to add some thoughts on this.

(2) In terms of getting a better understanding of the past experience of the males in these experiments, it would be nice to know how they were housed in the 2 weeks after activation onset (Line 88).

(3) I was a bit confused with the analyses. As I understand it, each male underwent 10 tests – 9 treatments and 1 control. The analyses, however, don't appear to take individual male into account (e.g., the analysis doesn't appear to be a repeated measure analysis). Unless I am misunderstanding, instead it appears as though there were 10 pairwise tests done; with no correction for multiple comparisons. It may well be that I am not fully understanding the custom analyses. Either way, it would be good to clarify. If the individual male's ID is not incorporated into currently analyses, I suggest revisiting the statistical approach.

Reviewer #2

(Remarks to the Author)

This manuscript reports on a series of experiments the authors designed to explore how animals that communicate by use of vibrational waves, which travel through the substrate on which they live, cope with noise, especially with human-created anthropogenic noise. Much has been learned in the past 2-3 decades about which organisms can use information contained in non-sound vibrational waveforms, including those produced by their species in courtship and group living. A general finding has been that if we test the organism, we find that they do use information in these substrate vibrations for things like locating each other, and courtship, since most animals are arthropods. We now know that hundreds of thousands of arthropod species communicate via substrate-borne vibrations, many using only these vibrations. The current state of our knowledge of the impact of noise on the biotic environment is based mostly on studies of sound (compressional waves detected by some kind of ear) and mostly only with vertebrate animals. Thus, this manuscript makes a major contribution of new knowledge to our understanding of noise in the biotic environment through rather elegant experiments that are focused on filling in gaps and expanding the scope of the questions while testing an insect that communicates with non-sound vibrations.

This very well written manuscript has real promise for becoming a classic paper. It should appeal to a broad readership

across fields. The authors chose a species well-known to them, the green stink bug, *Nezara viridula*, which has spread throughout terrestrial communities world-wide within recent years and established themselves as pests of dominant crops humans depend on for food. While researchers have been building a body of knowledge to use in combating these pests, there are only so many hypotheses that can be addressed at the same time, and knowledge gaps do exist. These authors chose to build on the vast amount of knowledge they do have about how green stink bugs use vibrations propagating through the substrates as they explore basic questions about how noise does or does not affect efficient signaling during courtship and mate location. They expanded on previous noise studies by testing with non-overlapping frequencies in the noise, as well as frequencies that did overlap the stink bug's signals. They used continuous white noise of different frequency bands and amplitudes to reveal any effects on sensory processing and behavior, then used electrophysiology techniques to check for signal representation in the leg nerves and even modelled their findings to estimate the effect of noise on frequency encoding of the Female Calling Song (the signal). The research team sought to improve their ability to predict long-term consequences of anthropogenic vibrational noise on this species by increasing their understanding of mechanisms at play through a carefully planned and executed experimental design. The conclusions are strong and fully supported by the findings of the paper. The findings can be used in future planning for humans as they build infrastructure, explore energy options, try to protect endangered lifeforms, deal with pest populations, etc.

Consider what we know about decisions made in the early 20th century based on incomplete knowledge that have left us with massive problems that persist today. Also consider how we are seeking cleaner, cheaper energy options while climates are becoming unbearably hot by moving into previously unexploited habitats, for example building giant structures to harness wind in our prairies and offshore without taking the time to strategize on what impacts these might have on the species that evolved in those habitats. This manuscript is important because of the scope of the study but also the unique aspects of its experimental design, the pioneering nature of the investigation and its research tools, and its timeliness in the present but also the future of our planet. Most humans are aware of the rapid increase in environmental noise in our own lives. Some are more affected than others, who either tolerate the noise or accept it as a by-product of economic advances. Yet, while in the recent decades urban planners are instituting new rules of how and where to construct new highways or airports, or requirements for safety equipment for use by human workers, etc., very little is really known about the effects of environmental noise created by human activities (anthropogenic noise) on non-humans or even non-vertebrate animals. This paper could serve as a model for asking and attempting to answer questions to address this lack of knowledge, even if these authors thought they were working to gain knowledge they needed to make predictions.

The paper is packed with information but presented in a highly readable form. The authors are thorough in explanations that make the text easy to follow. The figures add extra support for the text, supplying detailed findings, and are quite informative and well-drawn. The statistical analyses are state of the art and based on a wealth of detail from experimental design and findings. All in all, this paper can be used as a primer for those wishing to conduct similar experiments on their own species of interest.

Here are a few suggested edits:

Throughout, decide whether to use and or & in citations. Both are used, but changing is a quick fix by doing a word search. Just be consistent. These are in the Introduction, Conclusions and Line 465, but also in the References list.

Line 34: Cocroft and Rodriguez 2005

Line 47: such a situation

Line 64: parameters, while

Line 78: Zou instead of Zhou

Line 93: duet, prompting

Lines 110-111: bands comprised the following: fundamental-overlapping

Line 189: head of the vibration

Line 190: and noise as in

Line 215: simulate a sensory

Line 239: treatments; (Fig.

Lines 270 and 284: I am not sure about lines 270 and 284. In 270 you use "a" representative summed receptor neuron response, but do not in line 284. I am not sure if you mean one neuron's response or multiple neurons' responses. Please check and clarify this for future readers. Should they not match?

Line 364: female signal and the ability

Lines 380-381: another piece of evidence

Lines 396-397: receptor neurons (MFR), the only known leg receptor unit

Line 398: Summaric is a terrific modern word, but one this reviewer did not recognize. I am inclined to recommend you leave it in the text because it is specific to your description, but not everyone will grasp your point...unless, like me, they bother to look for it online! I am impressed, nonetheless!

Line 399: activity of predominantly low-frequency

Line 458: capacity for vibrational

Lines 483 and 489: Cocroft and Rodriguez 2005

Line 495: vibrational systems due

Line 499: interactions,

Line 500: understanding of the

Citations not on References list:

Line 73: Čokl 2008

Line 98: Michel et al. 1983; Čokl et al. 2006

Line 168: Mazzoni et al 2015

References list:

Lines 544-545: Not cited in the text? Could this be miscited as 2006, which was cited in the text on line 98 with no corresponding reference entry? This entry for 2007 is correct for an actual article, but not cited.

Line 557: 2021, not 2010

Line 576: 2015, not 2014

Line 578: Mazzoni, V., Gordon, S.D., Nieri, R., and Krugner R. instead of et al.

Lines 631-633: Is this entry out of alphabetical order in the References list?

Line 640: In the title, change to lower case instead of capitals to match other entries.

Reviewer #3

(Remarks to the Author)

This well written manuscript presents a set of experiments and modeling that test the effect of noise on communication with plant-borne vibrational signals in *Nezara* stink bugs. The experimental design is elegant and the dissection of the effect of noise that does or does not overlap the bugs' signals is novel and of broad interest to researchers in communication, behavior, physiology, and evolution. I do, however, have a couple of suggestions and questions regarding data analysis that I bring up with the goal of strengthening the impact of the manuscript.

1.
Each male was presented with a random sequence of all the stimuli in the playback experiment (L159-161) and in the physiology experiment (L194-198). I think it makes sense to get as much data out of each test individual as possible. However, this means that data across treatments and stimuli contributed by each male are not fully independent of each other, giving the experiments an element of pseudoreplication. The current statistical approach in the manuscript does not address this problem, and instead only accounts for multiple testing. I would suggest using statistical models that include male identity as a random term (together with fixed terms for treatment, SNR and their interaction for the playback experiment, for instance). This would allow full use of all the data while eliminating the problem of pseudoreplication.

2.
I did not fully understand the results in Fig. 5b. The solid lines indicate data during presentation of only noise. But the spike rate increases towards 0dB, which in this case means no female signal playback AND weaker noise ending in no noise at 0dB, correct? How can that increase neural response?

Minor:

L78:

I couldn't find Zhou et al. in References.

L106-107:

This could well be a feature of the physics of sound, as the authors interpret. But it could also arise from activation of the playback set up by the playbacks. I would suggest providing information about how the set up was isolated from building vibrations.

L128:

Ipsilateral relative to what?

L174-176:

Are non-participating males included in the % signaling data?

Reviewer #4

(Remarks to the Author)

The authors investigate the impact of noise on signal recognition and localization in the vibratory system of a stink bug. Specifically, they examine how the spectral overlap between signal and noise affects behavioral performance. Both spectrally overlapping and non-overlapping noise affected recognition and localization. Using recordings of the bulk activity of leg sensory neurons, they show that non-overlapping noise affects signal presentation the least, aligning with their behavioral results. Lastly, a simple model suggests that noise impairs localization by disturbing spike latencies between both legs.

Overall, this is an interesting contribution to the field. The paper is well-written, and the data analyses, modeling, and statistics are sound. I have only a few comments that I am sure the authors will be able to address:

1. The behavioral experiments use different ranges of noise than the physiology (-6 to 6 dB vs. 0 to 24 dB). In particular, the lowest SNR of -6 dB was not tested in the physiology. I found this surprising. Was this a limitation of the stimulation system used for physiology?

The authors state that "Extension to higher SNR levels than in behavioural experiments was used to determine the sensory threshold of noise effects." However, an explanation for why lower SNRs were not used is not provided. To be clear, I am not suggesting the authors perform any new experiments, but rather that the stimulus choice should be justified in Methods.

2. The authors compare behavioral data with their physiological and modeling results. However, this is never done quantitatively or graphically. I found it difficult to go back and forth between the figures showing behavioral data and comparing them to the physiology and modeling results. Could the authors present their data in a way that makes this comparison easier for the reader?

For instance, the relative PSD in Fig. 5c could be taken as a proxy for the male's signaling probability (Fig. 1a) and compared visually.

Similarly, the probability of correct localizations could be extracted from the data shown in Fig. 7 and compared graphically to the behavioral data on localization behavior.

If these suggestions are not feasible, perhaps the authors can devise alternative ways of better linking their different experiments and analyses.

This will show at least a qualitative match between behavior and physiology/modeling.

Version 1:

Reviewer comments:

Reviewer #1

(Remarks to the Author)

The authors did an excellent job revising this manuscript according to all reviewer's comments and I have no further concerns.

Reviewer #2

(Remarks to the Author)

Reviewer #3

(Remarks to the Author)

The authors have dealt well with the reviewers' comments, and I have no further suggestions for this interesting paper.

Reviewer #4

(Remarks to the Author)

The authors have addressed all my concerns.

In particular, I appreciate the new graphical summary in Fig. 8.

Below, we provide detailed responses to each of the comments, outlining the changes made.

In addition to suggested edits, we made a few additional ones correcting minor inconsistencies in our expressions. Notably, Figure 5 is altered after correcting an error in the formula for calculating standard errors, which resulted in slightly shorter error bars. This modification only concerns the graphics, and did not influence the results of underlying statistical tests or the interpretation.

Cordially,
the authors

Reviewer #1

(1) In reading recent papers on this topic, I found it noteworthy that the vibroscape can encompass both substrate-borne, but also airborne sound (e.g., Choi et al., Communication Biology 2024; Pessman et al., Entomologia Experimentalis et Applicata 2024). This seems especially relevant to the story that is being told here about the impact of non-overlapping noise. If there is space, it would be good to add some thoughts on this.

We acknowledge that this aspect is relevant, and have added a short mention in the discussion (lines 518-522). It does in fact emphasise the ecological relevance of the results.

(2) In terms of getting a better understanding of the past experience of the males in these experiments, it would be nice to know how they were housed in the 2 weeks after activation onset (Line 88).

Thank you for your comment. We agree that understanding the experience of the males is important for interpreting the results. We have added an explanation of how the males were housed after activation to lines 89-90 of the revised manuscript.

(3) I was a bit confused with the analyses. As I understand it, each male underwent 10 tests – 9 treatments and 1 control. The analyses, however, don't appear to take individual male into account (e.g., the analysis doesn't appear to be a repeated measure analysis). Unless I am misunderstanding, instead it appears as though there were 10 pairwise tests done; with no correction for multiple comparisons. It may well be that I am not fully understanding the custom analyses. Either way, it would be good to clarify. If the individual male's ID is not incorporated into currently analyses, I suggest revisiting the statistical approach.

Thank you for bringing this to our attention. We apologize for the confusion regarding the statistical analysis. You are correct in noting that the Wilcoxon rank sum test does not account for repeated measures. However, this was an error in reporting. The actual test performed was the pairwise Wilcoxon signed rank test, which appropriately accounts for the repeated nature of the experiment by incorporating individual male IDs. Additionally, we applied FDR correction for multiple comparisons to minimize the risk of type 1 errors. We have now corrected the reference to the statistical test used. The data and R code used to perform the tests can be verified in the supplemental material that was sent in with the submission. Additionally, we have also fitted linear mixed effect models to both behavioural and neurophysiological data, with treatment as a fixed effect and male ID as random effect. We added the details of the analysis to the Supplement. The post hoc tests show that the differences to previously used nonparametric tests are minor, indicating that male ID doesn't influence the statistical significance as much, so we have decided to keep the significance levels in the article as is. We apologize again for the oversight and appreciate your careful review.

Reviewer #2

Here are a few suggested edits: [...]

We have implemented all the suggestions to improve accuracy and clarity.

Below, we provide additional clarification to more specific concerns:

Lines 270 and 284: I am not sure about lines 270 and 284. In 270 you use “a” representative summed receptor neuron response, but do not in line 284. I am not sure if you mean one neuron’s response or multiple neurons’ responses. Please check and clarify this for future readers. Should they not match?

We would like to clarify that in both instances, the leg nerve response represents the activity of multiple receptor neurons. Figure 4 illustrates PSTH diagrams and raster plots derived from a single nerve recording that we identified as representative for all 12 recordings and Figure 5 presents data from all 12 recordings. To ensure consistency and clarity, we have adjusted the wording in these figure legends to clearly indicate that both instances refer to the summed response of multiple neurons, and added the definition to the glossary section of the supplementary file.

Line 398: Summaric is a terrific modern word, but one this reviewer did not recognize. I am inclined to recommend you leave it in the text because it is specific to your description, but not everyone will grasp your point...unless, like me, they bother to look for it online! I am impressed, nonetheless!

In line with the abovementioned adjustment, we have nevertheless changed this word to “summed” for consistency.

References list:

Lines 544-545: Not cited in the text? Could this be miscited as 2006, which was cited in the text on line 98 with no corresponding reference entry? This entry for 2007 is correct for an actual article, but not cited.

Thank you for noting this. This reference pertains to an older version of the manuscript, and we have removed it from the list.

Reviewer #3

1. Each male was presented with a random sequence of all the stimuli in the playback experiment (L159-161) and in the physiology experiment (L194-198). I think it makes sense to get as much data out of each test individual as possible. However, this means that data across treatments and stimuli contributed by each male are not fully independent of each other, giving the experiments an element of pseudoreplication. The current statistical approach in the manuscript does not address this problem, and instead only accounts for multiple testing.

I would suggest using statistical models that include male identity as random term (together with fixed terms for treatment, SNR and their interaction for the playback experiment, for instance). This would allow full use of all the data while eliminating the problem of pseudoreplication.

We acknowledge that presenting each male with a random sequence of stimuli means that data across treatments and stimuli are not fully independent, which introduces an element of pseudoreplication.

As highlighted in our response to Reviewer 2, we initially reported using the Wilcoxon rank sum test incorrectly. The actual test conducted was the pairwise Wilcoxon signed rank test, which appropriately accounts for the repeated measures nature of the data by including individual male IDs. We have corrected the reference to the statistical test and uploaded all relevant data for verification in the Supplement. We apologize for any confusion this may have caused.

In addition, to address concerns about pseudoreplication more effectively, we have now fitted linear mixed-effects models to both the behavioral and neurophysiological data. These models include male identity as a random effect and treatment and SNR as fixed effects, as you suggested. The post hoc tests indicate that the differences compared to the previously used nonparametric tests are minor, suggesting that male ID does not influence the statistical significance to a large extent. Therefore, we have decided to retain the original significance levels in the article and have added details of the modeling analysis to the Supplement.

We hope that this addresses your concerns and thank you for your suggestion that has led us to refine our statistical approach.

2. I did not fully understand the results in Fig. 5b. The solid lines indicate data during presentation of only noise. But the spike rate increases towards 0dB, which in this case means no female signal playback AND weaker noise ending in no noise at 0dB, correct? How can that increase neural response?

The increase in spike rate from 24 dB to 0 dB is the result of the noise amplitude increase relative to the signal, which is kept at the constant amplitude of 1 mm/s. At 0 dB SNR, the noise and the female signal are at the same amplitude. We have clarified this aspect in the Methods section (lines 118, 200) and added a brief explanation in the legends to Figures 4 and 5 to ensure clarity.

L78:

I couldn't find Zhou et al. in References.

The correct reference is Zou and we have corrected it accordingly in the text.

L106-107:

This could well be a feature of the physics of sound, as the authors interpret. But it could also arise from activation of the playback set up by the playbacks. I would suggest providing information about how the set up was isolated from building vibrations.

We have added a brief explanation to the Methods section on how we prevented any unintended stimulation of the setup (lines 102-104). Besides, an additional sentence aligning our finding of plant-induced signal harmonics with previous research (lines 108-111) is now included.

L128:

Ipsilateral relative to what?

Ipsilateral and contralateral always refer to the side relative to the minishaker emitting FCS. We have included this clarification to line 132 of the revised manuscript.

L174-176:

Are non-participating males included in the % signaling data?

Yes, since all males responded with vibrational signals to FCS playback in control trials, we surmised that non-participation was a result of disruption by noise.

Reviewer #4

1. The behavioral experiments use different ranges of noise than the physiology (-6 to 6 dB vs. 0 to 24 dB). In particular, the lowest SNR of -6 dB was not tested in the physiology. I found this surprising. Was this a limitation of the stimulation system used for physiology? The authors state that "Extension to higher SNR levels than in behavioural experiments was used to determine the sensory threshold of noise effects." However, an explanation for why lower SNRs were not used is not provided. To be clear, I am not suggesting the authors perform any new experiments, but rather that the stimulus choice should be justified in Methods.

Thank you for highlighting this discrepancy. The difference in noise ranges between the behavioral experiments and the physiological recordings was indeed a limitation. We did not test the lowest SNR of -6dB in the physiology because the high noise levels (increased relative to the fixed signal amplitude) led to mechanical interference being picked up with a measuring electrode. This interference was spectrally similar to the neuronal response, making it difficult to filter out. Additionally, our attempts to isolate the electrode using Mu-metal shielding were unsuccessful. We have included a brief explanation to the Methods section (lines 202-204) to clarify the rationale behind our stimulus choices.

2. The authors compare behavioral data with their physiological and modeling results. However, this is never done quantitatively or graphically. I found it difficult to go back and forth between the figures showing behavioral data and comparing them to the physiology and modeling results. Could the authors present their data in a way that makes this comparison easier for the reader?

For instance, the relative PSD in Fig. 5c could be taken as a proxy for the male's signaling probability (Fig. 1a) and compared visually.

Similarly, the probability of correct localizations could be extracted from the data shown in Fig. 7 and compared graphically to the behavioral data on localization behavior.

If these suggestions are not feasible, perhaps the authors can devise alternative ways of better linking their different experiments and analyses.

This will show at least a qualitative match between behavior and physiology/modeling.

We recognize the importance of making the comparison between behavioral data, physiological results, and modeling outputs clearer. In response to this suggestion, we have added a new figure (Figure 8) to facilitate the comparison. The figure provides an integrated view of the results, allowing for qualitative comparisons across the different types of data.